

# Mangrove Forest against Dyke-break induced Tsunami in Rapidly Subsiding Coasts

Hiroshi Takagi[1], Takahito Mikami[2], Daisuke Fujii[1], Miguel Esteban[3]

[1]School of Environment and Society, Tokyo Institute of Technology, Tokyo, 152-8550, Japan
[2] Department of Civil and Environmental Engineering, Waseda University, Tokyo, 169-8555, Japan
[3] Graduate School of Frontier Sciences, The University of Tokyo, Chiba, 277-8563, Japan

*Correspondence to*: H. Takagi (takagi@ide.titech.ac.jp)

**Abstract.** Thin coastal dykes typically found in developing countries may suddenly collapse due to rapid land subsidence, material aging, sea-level rise, high wave attack or a collision with vessels. Such a failure could trigger a dam-break tsunami-type flood, or *dyke-break induced tsunami*, a possibility which has so far been overlooked in the field of coastal disaster science and management. To analyze the potential consequences of one such flooding event caused by a dyke failure, a detailed hydrodynamic model was constructed based on the authors' field surveys of a vulnerable coastal location in Jakarta, Indonesia. Under a 2-m land subsidence scenario –which is expected to take place in the study area after only 10 years, the model results show that the flood waters rapidly rise to a height of nearly 3 m, resembling the flooding pattern of earthquake-induced tsunamis. The depth–velocity product criteria suggests that many of the narrow pedestrian paths behind the dyke could experience strong flows, which are far greater than the safe limits that would allow pedestrian evacuation. A couple of alternative scenarios were also considered to investigate how such flood impacts could be mitigated by creating a mangrove belt in front of the dyke as an additional safety measure. The *dyke-break induced tsunamis*, which in many areas are far more likely than regular earthquake tsunamis, cannot be overlooked and thus should be considered in disaster management and urban planning along the coasts of many developing countries.

## 1 Introduction

The utilization of coastal areas has dramatically increased over the last century. The growing population of most countries has brought about an extensive conversion of natural coastal landscapes to agriculture, aquaculture, residential, and industrial usages. About 23% of the world's population now lives within 100 km from coastlines, and population densities in coastal areas are about three times larger than the global average (Small and Nicholls, 2003, United Nations, 2006, Valiela, 2006). In fact, many of the world's largest cities have developed close to the sea. Jakarta, the capital of Indonesia, is one of such coastal megacities, with a population exceeding 9.6 million (as of 2010) an covering a total land area of 662 km$^2$, plus approximately 2.5 million daily commuters from neighboring cities (Djaja et al., 2004; Firman et al., 2010). The population growth rate reached 1.39 %/year over the period of 2000 – 2010 (Central Board of Statistics, 2010), which has made Jakarta one of the most densely populated cities in the world, and is expected to reach 30 million inhabitants by 2030.





As a consequence of this rapid development Jakarta has been facing many urban development issues. Amongst these, land subsidence appears to have become particularly serious over the last couple of decades. This problem was clearly recognized as far back as 1978, when substantial cracks were found in buildings and a bridge in downtown Jakarta (Djaja et al., 2004), and subsidence rates along the coast have varied between 9.5 and 21.5 cm/year in the period between 2007 and 2009

(Chaussard et al., 2013). This problem has been mostly caused by the widespread practice of extracting water for industrial uses, and has led to severe damage to buildings and infrastructure, increases in the extent of flooded areas, the destruction of local ground water systems and an increase in saltwater intrusion (Braadbaart and Braadbaart, 1997; Ng et al., 2012). If groundwater abstraction continues at the current rate it has been estimated that Jakarta would sink a further 5-6 meters by 2100 (Brinkman, 2012). As the Jakarta Coastal Development Strategy (Ministry of Infrastructure and the Environment,

2012) urges, policy makers and city officials should prioritize solving the land-subsidence problem as soon as possible, which obviously requires finding alternative provisions of clean water for the city.

Land subsidence due to the extraction of groundwater is a problem that has been faced by many other megacities in Asia (Fig.1). However, the current subsidence rate in Jakarta appears to be the fastest among the historical series of cities in the region, which have all substantially slowed-down in recent years. If effective countermeasures to mitigate land subsidence

are not undertaken in Jakarta, future coastal or fluvial flooding events will likely cause greater damage than what is expected at present, especially given the increased exposure due to population growth and migration to coastal areas. Budiyono et al. (2016) points out that the largest driver influencing on future flood risk in this city is land subsidence, exceeding the other factors such as sea-level rise, changes in extreme precipitation, and land use change, based on the global climate models (GSMs) and a hydrology model.

The authors are also concerned that land subsidence will rapidly lead to a decrease in the stability of coastal dykes, which even at present appear to be too thin and fragile to permanently hold back sea water (Fig. 2). Given the current rate of subsidence it is clear that the height of the dyke will need to be raised to compensate and maintain communities behind them safe. However, it is feared that this type of fragile dyke could collapse at any moment due to the subsidence, sea-level rise, the aging of concrete, or a collision with vessels. This would inevitably cause what the authors have termed a "dyke-break

induced tsunami", which would rapidly engulf the densely populated low-lying areas and likely cause massive loss of life.

The authors coined the term *dyke-break induced tsunami* (or *dyke-break tsunami*) to clearly illustrate to members of the public the danger and phenomenon that could be caused by the rupture of a coastal dyke. Local people are often unaware of the dangers posed by violent inundation events, such as for example what happened during the storm surge by the super strong typhoon Haiyan in 2013. After the event, many people expressed the view that it would have been better for

authorities and media to describe it by a simpler vocabulary such as a "tsunami" (Esteban et al., 2015, 2016, Mikami et al., 2016).

The authors consider that the risk of *dyke-break tsunamis* is far greater than regular earthquake tsunamis in Jakarta. Indeed, Indonesia is located in a tectonically active area where different plates (the Eurasian, Indo-Australian, Pacific, and Philippine Sea Plates) converge. Along the plate boundaries, there have been many earthquakes and volcanic activities. Some of the



earthquakes and volcanic activities generated tsunamis, which caused severe damage to coastal areas in Indonesia many times (Latief et al., 2000; Rynn, 2002; Brune et al., 2010; Paris et al., 2014). However, since Indonesia is composed of many islands and surrounded by different oceans and seas, each coastal area has different frequency and magnitude of tsunamis. The north of Jakarta faces Java Sea, which are surrounded by Borneo and Java Islands. According to Horspool et al. (2014),

coastal areas along Java Sea including Jakarta have relatively a low possibility of experiencing a large tsunami, compared with other areas, such as Indian Ocean coasts of Sumatra and Java Islands and coastal areas along the Molucca Sea. This is because there have been few major earthquakes recorded in the Java Sea and also because tsunamis, even if generated, will be blocked by many islands and narrow straits.

In the present paper the authors will assess the consequences that the collapse of coastal dykes would have on such a

vulnerable area by means of a detailed flood simulation based on a recent topographical survey in Jakarta. As a potential relatively low-cost countermeasure the authors propose the plantation of mangrove forests in front of the thin concrete dyke to increase the resilience of the structure and reduce flow velocities and damage in the event of a break. The effectiveness of such mangrove forests will also be evaluated by the numerical model.

## 2 Methodology

This section describes the basic data for the flood simulation which were obtained by the field investigation. Essentially, the authors fear that a dyke such as the one in Pluit is structurally deficient and is unlikely to withstand the excessive water pressures imposed on it during extremely high tides. Thus, countermeasures such as the planting of mangroves in front of it would be beneficial to improve its stability and reduce the consequence of a potential failure. The numerical model used will be briefly explained later in the paper.

### 2.1 Field investigation


The authors conducted a series of field surveys along the coastline of Jakarta (Fig.3) in January 2015, May 2015, September 2015, and February 2016 in order to investigate the current situation of Jakarta's coast and identify the potential coastal hazards affecting local communities. After these surveys, the authors clearly identified Pluit District as one of the most vulnerable coastal communities in Jakarta, and carried out a precise topographical survey of the district (Fig. 3 (c)). This area

is particularly important to flood protection in the city as it has a pump station, Pluit Pump Station, which is the largest in Jakarta, with a discharge capacity of nearly 50 m/s (JICA, 2013). Thus, the study of the area is not only important from the point of view of the consequences that a rupture would have for the local population, but the potential disruption for the Greater Jakarta due to the malfunction of this important facility.



## 2.2 Land subsidence scenarios

Coastal dykes in Jakarta need to be periodically heightened in order to keep up with the continuing land subsidence of the city, though this leads to a progressive decrease in structural stability. The dykes have already been overtopped, with Fig. 5 (left) showing the situation during the 2007 historical floods, and which were subsequently raised after the event. However, the dyke freeboard has almost disappeared after just 6 years (Fig. 5, right), demonstrating the rapid subsidence which has taken place in this period. In this sense, Chaussard et al. (2013) indicated that the rate of subsidence observed near Pluit District was 21.6 cm/year in the period between 2007 and 2009. As explained earlier, it is clear that effective countermeasures against land subsidence have not yet been put in place in Jakarta. Therefore, the authors have simply assumed that the current pace of land subsidence will continue in the future, with overall ground levels being 2 m lower than at present within the next 10 years. Although it is clear that this level of 2 m could be an overestimate if adequate policies are put in place, the authors consider that this is unlikely to happen, and thus flood simulations were performed for both present conditions and this future scenario, as illustrated in Fig. 6.

## 2.3 Mangrove plantation for mitigating floods

Indonesia has the world's largest proportion of mangrove surface area, though currently less than half its original extent (Global Nature Fund, 2007, Rasmeemasmuang and Sasaki, 2015). Mangroves usually grow in locations sheltered from waves and in areas between the mid-tide level and the highest high water spring tide. Within this intertidal zone, species have different preferences for elevation, salinity, and frequency of inundation (Global Nature Fund, 2007, Ellison, 2009).

One of the advantages of applying mangroves as a flood protection strategy is that they can be expected to naturally respond to increased sea levels or land subsidence and thus do not require expensive maintenance, unlike a concrete dyke system which has to be periodically upgraded to keep up with the pace of subsidence. Mangroves are considerably resilient with regards to protecting against shoreline change (Alongi, 2008). Gunderson et al. (2002) uses the term *ecological resilience* when talking about such abilities of ecosystems to recover from disturbance. Basically, accretion in these wetlands may be brought about by an accumulation of organic matter produced by the plants themselves (Valiela, 2006). In fact, available data on the piling-up rates suggest that many coastal marshes accrete at rates well comparable to sea level rise (e.g. Wood et al., 1989, Reed, 1990, Lynch et al., 1989, Day and Templet, 1989, Parkinson et al., 1994, Kearney et al., 1994).

*Avicennia marina*, commonly known as white or gray mangrove, is predominant in Jakarta and is particularly important to attenuate waves in the area (Oka et al., 2004, Herison et al., 2014). The Jakarta Bay is relatively calm in terms of its wave climate, and is also characterized by a high sediment flux from the many rivers in the area. Pluit District is particularly well sheltered from high waves, owing to the presence of two salient landfills (Fig. 3 (b)), which can be considered as an ideal location for mangrove rehabilitation. In fact, a mangrove rehabilitation project was already implemented at the Jakarta Fishing Port nearby Pluit (Fig. 3 (b)) and the effectiveness to reduce waves and stabilize the port breakwater have been acknowledged (Oka et al., 2004).



Another advantage of mangroves as a flood protection strategy is that they would effectively absorb wave energy or tidal current. The present study thus also analyses scenarios where a 20-m mangrove belt would be created in front of the concrete dyke of Pluit, which would essentially be similar to other existing mangrove protection in the vicinity (Oka et al., 2004). However, the extent of energy absorption by mangroves largely depends on tree density, diameter, bed slope, bathymetry,

characteristics of waves, and tidal condition (Alongi, 2008, Rasmeemasmuang and Sasaki, 2015). Thus, it is clear that more detailed analysis would be required to identify the ideal width of the mangrove belt for this particular site, though that is outside the scope of the present paper.

## 2.4 Numerical model and settings

Delft3D-FLOW was used to simulate a flood which rushes into the low-lying area of Pluit after a hypothetical break in the

dyke. In combination with a flooding scheme for advection in the momentum equation, the algorithm referred to as *drying and flooding* in this numerical model is expected to give a good representation of rapidly varying flows with large water-level gradients as a result of dam breaks (Stelling et al., 1986, Stelling and Duinmeijer, 2003, Deltares, 2011).

The simulation domain encompasses an area that includes Pluit District. A fine computational grid size (50 cm wide) was used throughout the domain, with 844,004 nodes reproducing small pedestrian paths with widths as narrow as 2 m. The

topography data were created based on the authors' own recent measurements. Although the length of the initial breach in the dyke and how this would propagate to nearby sections is unpredictable, the present study assumed a failure length of 20 m would occur, as shown in Fig.7. Then, the dyke-break induced flow was reproduced by imposing a constant sea level throughout the simulation, which ran for 20 minutes of real time. Velocity and inundation depth changes with time were plotted for 7 locations along several small paths amongst the houses in the area.

One of the most important numerical settings in any flooding simulation is Manning's $n$ value, particularly when simulating an urban area with many buildings. The present simulation assumes a uniform value of 0.06 over the land area, according to previous studies (Mignot et al., 2006, MLIT, 2012, Takagi et al., 2016). It is noted that a larger Manning's $n$ value may need to be used when a flood causes catastrophic damage to buildings over a wide area (Koshimura et al, 2009, Takagi and Bricker, 2014, Bricker et al., 2015). However, extensive destruction of buildings is not expected as a consequence of a

limited dyke failure, as discussed in the following section, and thus a moderate $n$ value appears reasonable.

Although there is some uncertainty in the $n$ number that should be used for mangroves, a value of 0.15 was used in this study, according to the work of Arcement and Schneider (1984), in which the $n$ value was assumed to lie between 0.10 and 0.20 for extremely dense vegetation. Since mangroves grow in the range between mid-tide level and the highest high water spring tide, the water depth in the mangrove area was assumed to be 50 cm (considering also the tidal range in Jakarta Bay).

Although the present study uses Manning's $n$ value for the representation of the resistance to the water flow due to the mangrove roots, it should also be noted that other parameters such as drag coefficients related to Reynolds number, plant length, diameter, and vegetation density would have to be taken into account (e.g. Mazda et al. 2006, Mendez and Losada,





2004, Losada, 2016). These effects are clearly important and should be incorporated into more sophisticated CFD models in the future study.

## 3 Results and Discussion

Fig. 8 presents time variations of the flood in terms of flood depth, velocity, and their product, referred to as the depth-velocity product, for the 4 scenarios described in the preceding sections (namely representing the present or future elevations, and whether a mangrove belt in front of the dyke is considered for each of them). All the results indicate that the floodwater would arrive to the small paths a few minutes after the breach, covering the entire town within 10 minutes. This clearly demonstrates that land subsidence inevitably poses a great flood risk, and that this will substantially increase as subsidence progresses. It is also noted that rise in water level is not gradual, but rather abrupt, resembling which resembles the pattern of earthquake-induced tsunamis.

A large sized water flume experiment in Japan found that a 60 year old man did not experience any strong feelings of fear when he walked against a water flow as high as his crotch with speeds up to 0.8 m/s (Suga et al., 1995). However, he felt intense fear when the flow reached a height of 1 m or a velocity of 1 m/s, which effectively prevented him from walking without a supporting rope. Nishihata et al. (2005) confirmed through the same type of experiments that the maximum inundation depth at which adult males and females (aged from 21 to 59) can safely evacuate is 0.7 m or less. Takagi et al. (2016) also found that the family members featured in a video taken during the storm surge caused by Typhoon Haiyan which hit the Philippines in 2013 could cross a flooded street because the flow speed and depth were both relatively small, 0.6m/s and 0.6 m, respectively. Wright et al. (2010) proposes a depth-velocity product of 1.0m$^2$/s as the safe limit for pedestrians. Given these criteria pedestrians would be at great risk over the entire area, especially for the scenarios that do not consider the mangrove belt in front of the dykes, as shown in Figs. 8 (a) and (c). On the other hand, the simulations show that mangroves could substantially reduce the physical impact of the flood and enable pedestrians to evacuate to safer locations, as shown in Figs. 8 (b) and (d).

The depth-velocity product can also be used as a criteria to investigate whether buildings can collapse or not. Residential house damage can be determined by the depth-velocity product (dv) by following the classification given by Pistrika et al., (2010):

- *inundation damage*  (dv < 3m$^2$/s)
- *partial damage*  (3m$^2$/s $\leqq$ dv < 7m$^2$/s)
- *total destruction*  (dv $\geqq$ 7m$^2$/s)

According to this classification it is clear that houses can withstand much stronger floodwaters than pedestrians, as expected. Fig. 8 shows how it is unlikely that any of the buildings in the area would suffer any structural damage, given that depth-velocity products would generally be below 3m$^2$/s. Thus in the event of any dyke failure the residents should immediately



evacuate to any house in the area that is 2 stories or higher, rather than attempt to follow the narrow paths to high ground or another area.

The two scenarios with a mangrove belt suggest that both flood depth and velocity would be significantly reduced by their presence, compared with the cases without such countermeasures. It is also interesting to note that mangroves may

contribute to the smoothening of the flood stream by creating a large friction due to their dense root systems. Essentially, a sudden dyke-break would produce substantial waves in addition to uniform floodwater, if the mangroves are not present, as shown in Fig. 8 (c). Even though the flood depth is not different, walking in almost uniform flow would probably be easier than in a strong turbulent flow. Thus, it is clear that developing a mangrove belt can significantly improve the resilience of vulnerable coastal communities.

One important difference between the two types of tsunamis mentioned in this paper, namely the *dyke-break tsunami* and *earthquake-induced tsunami*, is that a flood caused by the former event will remain over the ground for a long duration, whereas an earthquake tsunami typically recedes in a relatively short period due to backwash. Thus, the event of dyke-break is not only a matter for local community, but would have a significant effect over a much wider region. As mentioned earlier, the largest pump station in Jakarta is located in Pluit District, and has been playing a vital role to discharge the storm water

that accumulates over the Jakarta metropolitan area into the bay. In case that a dyke-break tsunami occurs, floodwater would remain over a long period of time and consequently the pump system might be crippled, resulting in the malfunction of economic activities, people's daily life and traffic system in the Great Jakarta region. Thus, it should be recognized that the protection of this local dyke is not only a local issue but also essential for flood control in the entire city, which is of great importance to the entire country given the importance of the area to the national economy.

**4 Conclusions**

In the present paper the authors coined the term "dyke-break induced tsunami (or, simply dyke-break tsunami)", and attempted to highlight the dangers and extensive damage that such an event could cause to extremely low-lying areas in Indonesia. For this purpose, a detailed case study was conducted on the Pluit District in Jakarta, where the densely-populated community has been suffered rapid rates of land subsidence, in excess of 20 cm/year. The hydrodynamic simulation,

conducted using a very fine computational grid, estimated that the flood water would engulf the populated area through its narrow paths in a question of a few minutes. In the 2-m subsidence scenario, expected to be reached within the next 10 years, the depth-velocity products exceed the pedestrian evacuation limit of 1 $m^2$/s all throughout the town, and could seriously endanger any resident in the streets.

This study also demonstrates that the impact of the flood would be substantially mitigated by planting a mangrove belt in

front of the dyke, due to two mechanisms: (1) a reduction in floodwater velocity and depth and (2) flow smoothening effect, which mitigates the danger of turbulence flow. The effectiveness of mangroves is expected to become more pronounced with the progress of land subsidence. Although this study only deals with one small community in Jakarta, any developing countries or coasts are likely to encounter similar issue when faced with rapid industrialization and population growth. Thus,



the risk of *dyke-break tsunamis* should be considered in coastal disaster management and urban planning, and this study points out how planting mangroves could potentially be a good way to increase the resilience of these coastlines.

**Acknowledgements**

Funding for this research was supported by the Environment Research and Technology Development Fund (S-14) of the
Ministry of the Environment, Japan. The author thanks Hendra Achiari and Dhemi Harlan for their assistance during the preliminary field survey in Jakarta and JanJaap Brinkman and Junichi Fukushima for sharing their profound knowledge on Jakarta's flood issues.

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

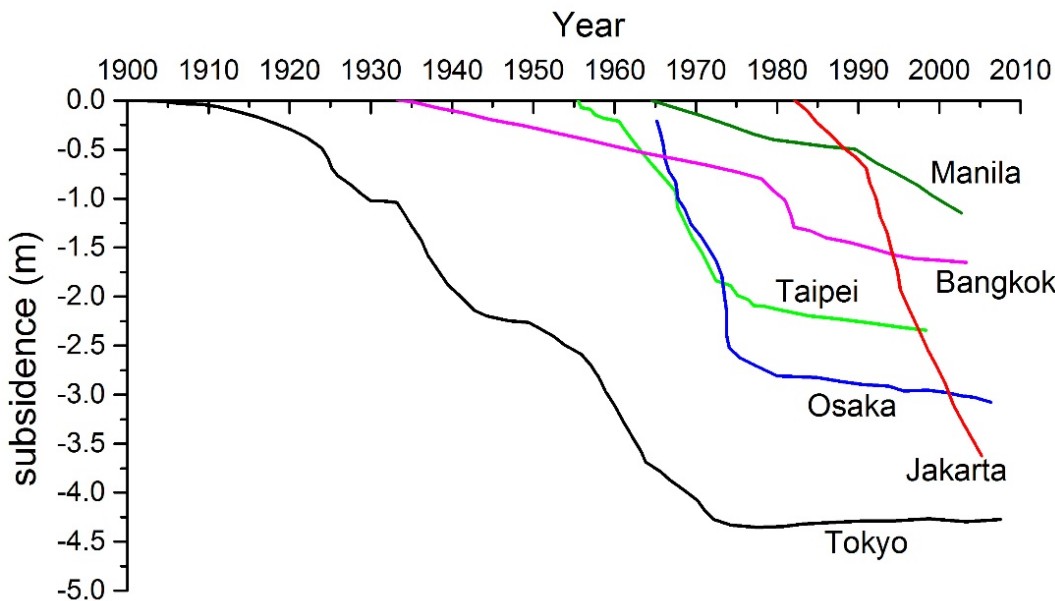

**Figure 1: Land subsidence in Asian megacities (drawn by the authors using data from Kaneko and Toyota, 2011).**






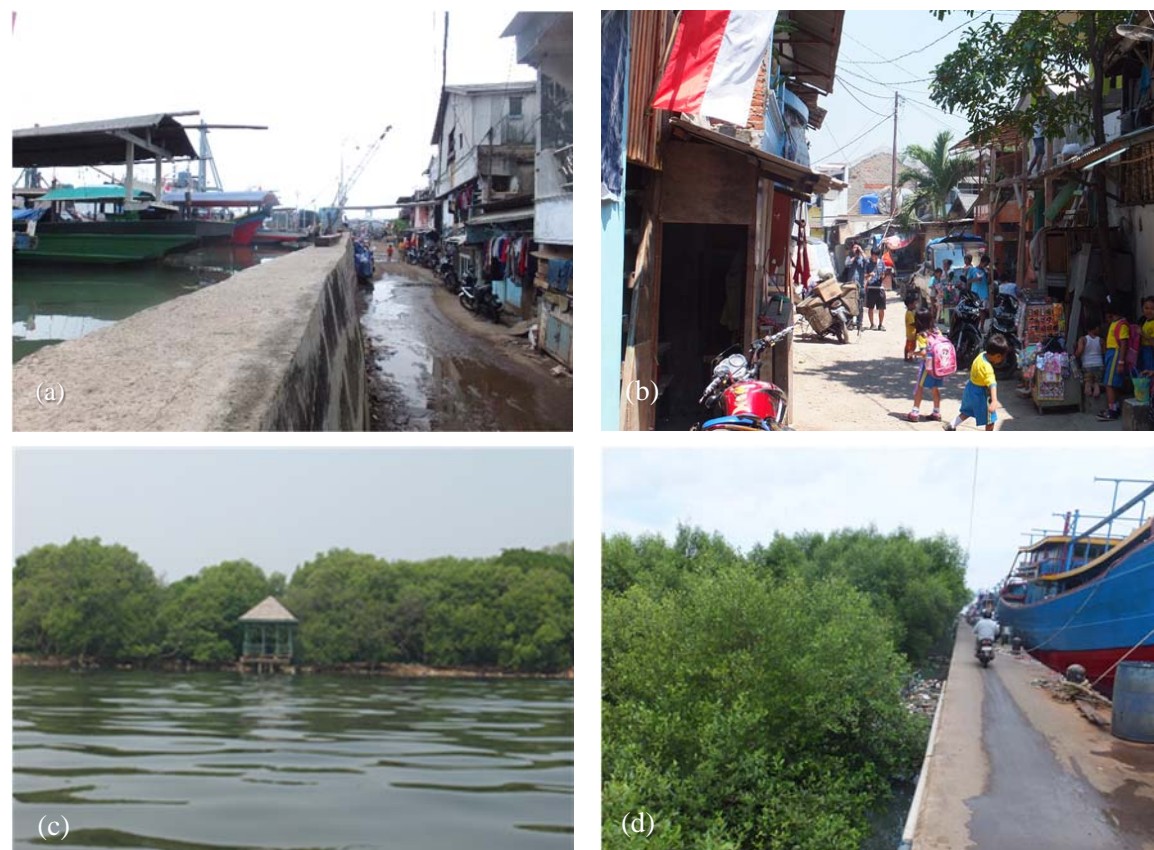

**Figure 2: Coastal areas around Jakarta (photos taken by the authors during field surveys in the period between May 2015 and February 2016) (a) A thin coastal dyke protecting a community experiencing rapid subsidence (Pluit District, northern Jakarta). Leakage of seawater through a concrete joint was observed during the authors' visual inspection on February 6, 2016. The sea level was 1.3 m higher than the road. (b) The town center of Pluit, with a high population density and long narrow paths. (c) Mangrove conservation site in Jakarta (see Fig. 3 on its location) (d) Mangrove rehabilitation site in the Jakarta Fishing Port (see Fig. 3 on its location). The mangrove belt is around 20 m thick and can effectively protect the port basin from wind waves.**





**Figure 3: (a) Map of central and northern Jakarta, (b) Study area, which is adjacent to Jakarta's largest fishing port. A yellow inverse triangle indicates the location of the installed pressure sensor, and (c) Pluit District, in which the authors measured ground elevations with a laser rangefinder (Fig.2 (b)). The ground elevations were adjusted to the highest water level shown in Fig. 4.**





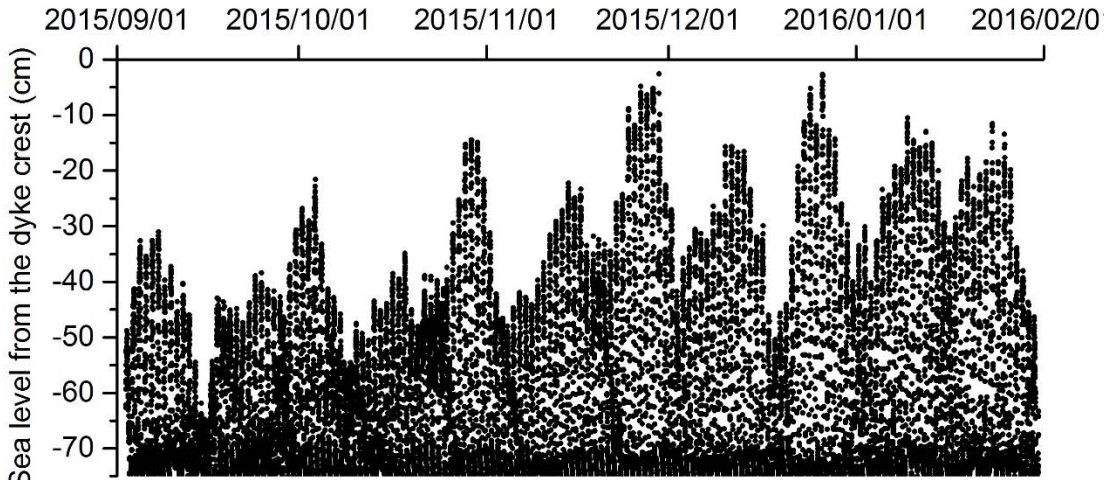

**Figure 4: Sea level in Pluit bay with reference to the top of the dyke (Fig.3 (b)). Note that the top of the graph represents the top level of the dyke, and the y-axis represents the maximum water level with reference to that point. Sea levels were recorded at 10-min intervals between September 2015 to February 2016, and were derived by subtracting atmospheric pressures from total pressures.**

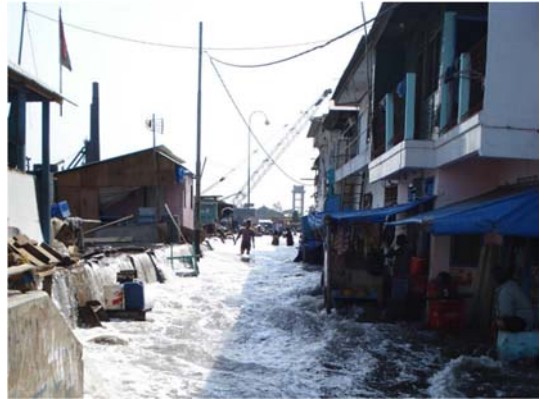 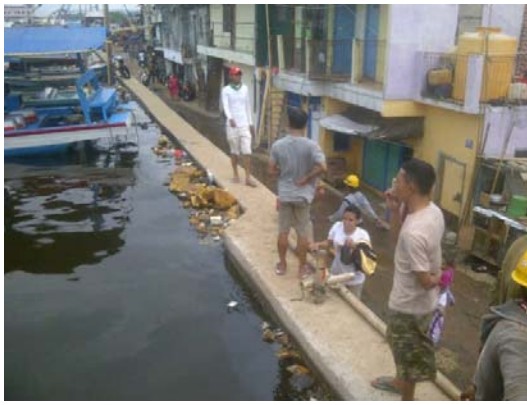

**Figure 5: Pluit District in Jakarta is situated below sea level. The area suffered extensive inundation during a high tide on November 26, 2007 (Left). The thin dyke protecting the settlement was raised by about a meter after the 2007 event by the local government, obstructing people's vision of the sea. The photo was taken on October 18, 2013 (Right). (Photos: courtesy of**

10 **Brinkman J.J., Deltares) .**




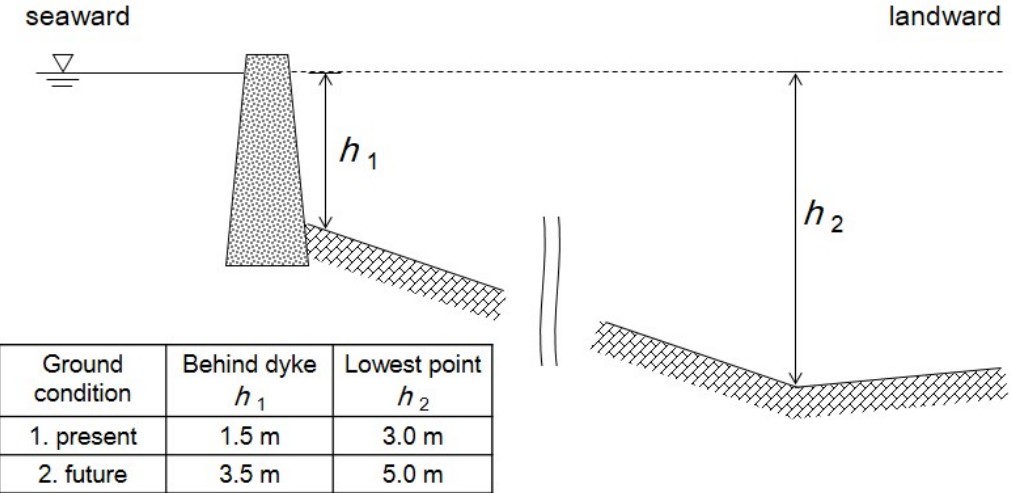

**Figure 6: Schematic view of the topography considered in the numerical simulation. Scenario 1: present topography constructed based on the authors' recent surveys, Scenario 2: future topography after experiencing an additional two-meter land subsidence. $h_1$ denotes the height from the water level behind the dyke, $h_2$ the height at the lowest location in the Pluit District.**





**Figure 7: Elevation topography used in the simulation, measured from the sea level: (a) Present ground elevations and buildings (dark green), (b) Present elevations, together with a 20-m mangrove belt in front of the dyke. Notes: The enlarged view focuses on the dyke-break section, assumed to be 20 m long. The numbers 1 through to 7 indicate the output points of the computational results.**



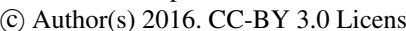



**Figure 8: Simulated water depth (left), velocity (center), and depth – velocity product (right) at 7 locations along the small paths shown in Fig. 7: (a) present elevations, which was reproduced based on the authors' survey, (b) present elevations with a 20-m mangrove belt, (c) 2-m land subsidence scenario, and (d) 2-m land subsidence scenario with a 20-m mangrove belt.**