# Peer review of "Mangrove Forest against Dyke-break induced Tsunami in Rapidly Subsiding Coasts"

_Natural Hazards and Earth System Sciences, 2016_

## Referee Comment (RC1) · S. D. ROSEN (Referee) · 15 Jun 2016

Review of nhess-2016-128, 2016 paper titled "Mangrove Forest against Dyke-break induced Tsunami in Rapidly Subsiding Coasts" by Hiroshi Takagi, Takahito Mikami, Daisuke Fujii, Miguel Esteban

a. General Comments and Remarks The article of Hiroshi Takagi and his collaborators approaches an important subject, namely the risk associated with subsiding coastal sites under potential flooding of a type similar to that induced by tsunamis. The authors investigated thoroughly a coastal section presently protected by thin coastal dykes and bring a potential relatively cheap solution for reducing the flooding risk to the local population via plantation of mangroves. While the proposed solution may be adequate for a temporary protection of a number of 10-20 years, we believe that it will

not provide protection in the long run under the foreseen climate change induced global sea level rise. A number of specific remarks are presented below and in section b in Table 1 is provided a list of technical and typographical corrections sugestions to the article contents. 1. The nick-naming of the dyke-break induced flooding as tsunami for greater awareness of public is understandable, but because it is misleading due to its prolonged flooding, in this reviewer's opinion, it should not be accepted. Instead, a plain nick name such as "dyke-break extreme flooding"or at least "dyke-break induced tsunami like flooding" would be preferable. If my opinion is accepted all terms in the text should be corrected accordingly. 2. It would be advisable that the authors mentioned sea water desalination as a counter action potential solution against land subsidence induced by underground water withdrawal. 3. A fast and significant subsidence rate has been indicated for the recent past years. It is not clear on what basis the same rate is maintained for the coming 10 years as well as for further time states. The subsidence would depend on the soil type of the underground and the thickness of the pervious layers, so it is not necessary correct to extrapolate the same sinking rate for the future, unless the pervious soil and its thickness give base to this assumption, fact that is not stated. 4. It is not clear also if the plantation of mangrove forest will be able to provide the expected protection in future. The present water depth in the proposed plantation area is indicated as 50 cm and that the plants grow at approximately as the present sea level rise. However, new publications (e.g. Dutton et al., 2015; DeConto and Pollard, 2016; Hansen et al., 2015; Mengel et al., 2015;) indicate a feasible faster and larger sea level rise globally (up to 1m by 2050, 2-3 m by 2100), in which case the mangrove will not be able to grow at the same rates and provide the expected protection from dyke-break rapid flooding. Perhaps an engineering alternative could be to adopt the Dutch concept, of building wide sand dunes at the waterline, requiring resettling of population and its activities back to higher and more remote places from the lower sea/water front areas. Another theoretical option might be migration to higher places (Roberts and Andrei, 2015) or that adopted by Miami City in USA (Weiss, 2016). 5. The criteria proposed following the classification given by Pistrika et al. as well as
the one proposed by Wright et al., (2010) seem very problematic as explained further below. Also the data brought by Suga et al., 1995) indicating a safe velocity limit of up to 0.8 m/s in a water depth of about 0.8 m, whereas at speed of 1 m/s was the highest safe limit walking against the current. Based on a research beach bathers survey study carried in Japan with in order to determine safe recreational conditions, that paper stated a safe limit of current speed of about 0.15m/s for knees deep water flow (about 0.5m depth), beyond which bathers could not walk normally or remain stable. Unfortunately, I was not able to find this article published in the "Coastal Engineering in Japan", in the 1980's. In the present article, the authors selected to use a depth velocity product criterion to determine safe passage of pedestrians in a flooded area. The present paper describes a flooding in the Philippines where people could cross a flooded street in a water depth of 0.6 m (knees depth) and while a 0.6 m/s current flow was present.

The information provided about the persons particulars is very limited and about which type of street (paved, unpaved, etc) was crossed in the flooded area. This seems already dubious as it is not clear how one was able at the time to measure the current speed, which, if it was 0.6 m/s (based on the Japanesse paper I mentioned), should have been done by a tall, heavy and strong person with perhaps even some cable support from being carried away. A velocity-depth product of 1.0 m2/s seems already unsafe, if we consider that this product can be due to various scenarios, such as: a depth of 1m and speed of 1.0 m/s (2 knots); a 1.8 water depth in a 0.55 m/s current (1 knot); or a water depth of 0.6m in a current speed of 1.67 m/s. These all lead to same velocity-depth product of 1 m2/s. A more rigorous approach is the work of Cox et al., 2010, quoted by Pistrika et al., which in this reviewer's opinion is a very important one. We believe it would be appropriate that the authors quote the following text taken from Cox and al., 2010, or at least refer to it and give the a summarizing figure from that publication, copied further below as Figure 1. Since the Cox et al. report is more recent and of broader coverage, and since it refers in greater detail to the various types of persons and ages and floor bottom conditions, even if the Japanesse article was

right for the wave induced current under open coast conditions with waves and sandy sea bottom, we estimate using the Cox et al. report would be more adequate for use in the present article.

"Significant scatter is observed within individual experimental data sets and, to a more significant degree, when all data sets are combined. Additionally, markedly differing tolerable D.V values are observed for identical subjects. Discussion with investigators has indicated that "training" of the subject (Abt, pers. com, 2009) may enable higher flow values to be resisted as the subject learns how to position the body so to best resist the flow. The lowest stability values (D.V) for each subject is, in most cases, the first exposure test and more applicable to the general population whom have not had the benefit of such training prior to encountering flood water. While distinct relationships exist between a subjects height and mass (H.M; mkg) and the tolerable flow value (D.V; m2s-1), definition of general flood flow safety guidelines according to this relation is not considered practical given the wide range in such characteristics within the population. In order to define safety limits which are applicable for all persons, hazard regimes are defined for adults (H.M > 50 mkg) and children (H.M = 25 to 50 mkg). Infants and very young children (H.M < 25 mkg) are considered unsafe in any flow without adult support. For children with a height and mass product (H.M) of between 25 and 50, low hazard exists for flow values of D.V < 0.4 m2s-1, with a maximum flow depth of 0.5 m regardless of velocity and a maximum velocity of 3.0 ms-1 at shallow depths. Under these flow regimes, the children tested retained their footing and felt "safe" in the flow. For adults (H.M > 50), low hazard exists for flow values of D.V < 0.6 m2s-1 with a maximum depth limit of 1.2 m and a maximum velocity of 3.0 ms-1 at shallow depths. Moderate hazard for adults exists between D.V = 0.6 to 0.8 m2s-1, with an upper working flow value of D.V < 0.8 m2s-1 recommended for trained safety workers or experienced and well equipped persons. Significant hazard for adults exists between D.V = 0.8 to 1.2 m2s-1. For flow values D.V > 1.2 m2s-1 the majority of tests for adults indicated instability - the hazard is extreme and should not be considered safe for standing or wading.

Figure 1 – Appropriate safety criteria for pedestrians walking in a flooded area Fig. 1

It should however be noted that loss of stability could occur in milder flow regimes when adverse conditions are encountered including: • Bottom conditions: uneven, slippery, obstacles; • Flow conditions: floating debris, low temperature, poor visibility, unsteady flow and flow aeration; • Human subject: standing or moving, experience and training, clothing and footwear, physical attributes additional to height and mass including muscular development and/or other disability, psychological factors; • Others: strong wind, poor lighting, definition of stability limit (i.e. feeling unsafe or complete loss of footing)."

b. Table 1 List of technical and typographical corrections sugestions to the article content Fig. 2

c. References listed by Rosen, excluding those in the referred article:

Cox, R. J., T. D. Shand, and M. J. Blacka, 2010. Australian rainfall and runoff revision Project 10: Appropriate safety criteria for people, Stage 1 report, April 2010, Engineers Australia Engineering House, Water Research Laboratory, The University of New South Wales

DeConto R., and D. Pollard, 2015. Modeling Antarctica's contribution to sea-level rise during the Last Interglacial and the future: differing roles of oceanic versus atmospheric warming, Geophysical Research Abstracts, Vol. 17, EGU2015-8104, 2015, EGU General Assembly 2015

DeConto R., and D. Pollard, 2016. Contribution of Antarctica to past and future sea-level rise, 31 March 2016, VOL 531, NATURE 591

Dutton A., A. E. Carlson, A. J. Long, G. A. Milne, P. U. Clark, R. DeConto, B. P. Horton, S. Rahmstorf, M. E. Raymo, 2015. Sea-level rise due to polar ice-sheet mass loss during past warm periods, Science 349, aaa4019 (2015). DOI: 10.1126/science.aaa4019

Hansen, J., M. Sato, P. Hearty, R Ruedy, M. Kelley, V. Mason-Delmotte, G. Russell,

[Figure]

G. Tselioudis, J. Cao, E. Rignot, I. Velicogna, E. Kandiano, K. von Schuckmann, P. Kharecha, A.N. Legrande, M. Bauer and K.-W. Lo., 2015. Ice melt, sea level rise and superstorms: evidence from paleoclimate data, climate modeling, and modern observations that 2o C global warming is highly dangerous, Atmos. Chem. Phys. Discuss, 15, 20059-20179, 2015.

Mengel, M., A. Levermann, K. Frieler, A. Robinson, B. Marzeion, and R. Winkelmann, 2015. Future sea level rise constrained by observations and long-term commitment, PNAS, March 8, 2016, vol. 113, no. 10, 2597–2602, www.pnas.org/cgi/doi/10.1073/pnas.1500515113

Roberts E., S. Andrei, 2015. The rising tide: migration as a response to loss and damage from sea level rise in vulnerable communities, Int. J. Global Warming, Vol. 8, No. 2, 2015

Weiss J., 2016. Miami Beach's 400 Million Sea-Level Rise Plan Is unprecedented, but Not Everyone Is Sold, The Miami New Times,Tuesday, April 19, 2016. http://www.miaminewtimes.com/news/miami-beachs-400-million-sea-level-rise-plan-is-unprecedented-but-not-everyone-is-sold-8398989

| DV (m²s⁻¹) | Infants, small children (H.M ≤ 25) and frail/older persons | Children (H.M = 25 to 50) | Adults (H.M > 50) |
|---|---|---|---|
| 0 | Safe | Safe | Safe |
| 0 – 0.4 | Extreme Hazard; Dangerous to all | Low Hazard[1] | Low Hazard[1] |
| 0.4 – 0.6 | | Significant Hazard; Dangerous to most | |
| 0.6 – 0.8 | | Extreme Hazard; Dangerous to all | Moderate Hazard; Dangerous to some[2] |
| 0.8 – 1.2 | | | Significant Hazard; Dangerous to most[3] |
| > 1.2 | | | Extreme Hazard; Dangerous to all |

[1] Stability uncompromised for persons within laboratory testing program at these flows (to maximum flow depth of 0.5 m for children and 1.2 m for adults and a maximum velocity of 3.0 ms⁻¹ at shallow depths).
[2] Working limit for trained safety workers or experienced and well equipped persons (D.V < 0.8 m²s⁻¹)
[3] Upper limit of stability observed during most investigations (D.V > 1.2 m²s⁻¹)

**Fig. 1.**

| item # | page # | line# | from | Existing text: | Suggested correction: |
|---|---|---|---|---|---|
| 01 | 1 | 1 | top | Dyke-break induced Tsunami | Dyke-break induced Tsunami like Flooding |
| 02 | 1 | 10 | top | type flood | Type flooding |
| 03 | 1 | 10 | top | induced tsunami, | induced tsunami like flooding |
| 04 | 1 | 18 | top | induced tsunamis, | induced tsunami like flooding events, |
| 05 | 1 | 19 | top | Tsunamis | tsunami events |
| 06 | 1 | 27 | top | of 2010) an | of 2010) and |
| 07 | 2 | 24 | top | aging of concrete | ageing of reinforced concrete ( *UK English)* |
| 08 | 2 | 31 | top | 2016). | 2016). However, since tsunami flooding is of short duration (hours to one day) while the dam-break induced tsunami like flooding is of a very long duration (days), the name "dyke break induced tsunami like flooding" is of selected to prevent confusion. |
| 09 | 3 | 25 | top | pump station | pumping station |
| 10 | 5 | 21 | top | value of 0.06 | value of 0.06 *(what units system since its value is dimensional dependent, i.e.* $[m^{(-1/3)} \cdot sec]$ or $[ft^{(-1/3)} \cdot sec]$ *??), for example:* value of 0.06 [SI] |
| 11 | 5 | 29 | top | tidal range in Jakarta Bay) | tidal range in Jakarta Bay, which is **???** cm) |
| 12 | 6 | 9 | top | resembling which resembles | which resembles |
| 13 | 7 | 10 | top | dyke-break tsunami | dyke-break induced tsunami like flooding |
| 14 | 7 | 11 | top | earthquake-induced tsunami | earthquake-induced tsunami flooding |
| 15 | 7 | 15 | top | dyke-break tsunami | dyke-break induced tsunami like flooding |
| 16 | 7 | 21 | top | dyke-break induced tsunami | dyke-break induced tsunami like flooding |
| 16 | 7 | 21 | top | simply dyke-break tsunami) | simply dyke-break flooding) |
| 16 | 7 | 31 | top | turbulence | turbulent |

**Fig. 2.**

---

## Referee Comment (RC2) · Anonymous Referee #2 · 21 Jun 2016

Revision comments to the paper: Mangrove Forest against Dyke-break induced Tsunami in Rapidly Subsiding Coasts. H. Takagi1, T. Mikami2, D. Fujii1, M. Esteban3 The paper is well written and documented, but there are a few misspellings Page Line Comment 2 5 Correct abstraction by extraction 2 23 Correct dyke by dykes 3 26 Correct m/s by m3/s 6 1 Correct Losada by Losada et al. 6 9 Change "resembling which resembles" by "like" 9 21 Reference Kaneko S., Toyota T. is not commented in the text 10 12 Reference Reed D.J. is not commented in the text

Figure 7 showing the models bathymetry indicate a steady topography descent from the upper part of the domain (where the dyke breach is located) to the opposite side. Despite the poor selection of colours it can be seem that the roads around the breach are at around -1.5 m, while point 7 and beyond in the other side of the domain is about -3 m or more. This is not congruent with the much complex measured topography

indicated in figure 3. The velocity figures for the present scenario without mangrove protection and in all cases with it show that by the simulation end (20 minutes) water is flowing at steady pace in all points. As water depth is also stabilised, where is the water flowing?. There are open boundaries?. There is insufficient data on the paper about the boundaries of the numerical domain, so results are of difficult interpretation based on the presented information. The wild oscillations of water level and velocity shown in the case of 2 m subsidence case without mangrove protection are not commented. Could it be related to model instabilities?. A better figure 7, showing clearly the model boundaries and the topography will be welcomed. In relation to the capacity of people to withstand a water flow of a given depth and velocity, the proposition of Wright et al. (2010) of a depth-velocity product of 1.0 m2/s as the safe limit for pedestrians seems optimistic. On that respect, Jonkman, S.N. and Penning-Rowsell, E. (2008) proposed some formulas for both moment and sliding instability of pedestrians, depending on the individual mass, friction factor and flow depth and velocity. Using these formulas, this 1 m2/s limit seems only valid for trained adults. A more recent paper of Cox et al. (2010) provides a much more realistic table about the safe limits for wadding in water flows for people.

References cited:

Cox, R.J., T.D. Shand and M.J. Blacka (2010). Australian Rainfall and Runoff. Revision Project 10: Appropriate Safety Criteria for People. Engineers Australia Engineering House, Water Research Laboratory, The University of New South Wales.

Jonkman, S.N. and Penning-Rowsell, E. (2008). Human Instability in Flood Flows. Journal of the American Water Resources Association, Vol. 44, No. 4, pp 1 – 11.

Please also note the supplement to this comment:
http://www.nat-hazards-earth-syst-sci-discuss.net/nhess-2016-128/nhess-2016-128-RC2-supplement.pdf

---

## Author Comment (AC1) · 26 Jun 2016

**Reply to the comments:**

In blue: reviewers' comments
In red: authors' reply

**Reviewer #1, Dr. S. D. Rosen:**

General Comments and Remarks

The article of Hiroshi Takagi and his collaborators approaches an important subject, namely the risk associated with subsiding coastal sites under potential flooding of a type similar to that induced by tsunamis. The authors investigated thoroughly a coastal section presently protected by thin coastal dykes and bring a potential relatively cheap solution for reducing the flooding risk to the local population via plantation of mangroves. While the proposed solution may be adequate for a temporary protection of a number of 10-20 years, we believe that it will not provide protection in the long run under the foreseen climate change induced global sea level rise.

A number of specific remarks are presented below and in section b in Table 1 is provided a list of technical and typographical corrections suggestions to the article contents.

We thank the reviewer for the great number of very productive comments and suggestions, which would enable us to significantly improve our manuscript.

1. The nick-naming of the dyke-break induced flooding as tsunami for greater awareness of public is understandable, but because it is misleading due to its prolonged flooding, in this reviewer's opinion, it should not be accepted. Instead, a plain nick name such as "dyke-break extreme flooding" or at least "dyke-break induced tsunami like flooding" would be preferable. If my opinion is accepted all terms in the text should be corrected accordingly.

As described in Lines 24-31, p.2, we coined the term *dyke-break induced tsunami* in order to clearly illustrate to members of the public the danger and phenomenon that could be caused by the rupture of a coastal dyke. Local people seem to be unaware of the dangers posed by this type of sudden violent flood events. For example super strong

typhoon Haiyan in 2013 caused a massive storm surge, claiming more than 6000 lives in the Philippines, even though the meteorological agency in the Philippines issued a typhoon warning with a potential *storm surge* height up to 7m a day before the landfall. According to the authors' post-disaster survey, however, a number of local inhabitants could not realize what would happen due to *storm surge* as many of them had only just heard the term for the first time. Many people expressed the view that it would have been better for authorities and media to describe it by a simpler vocabulary such as a *tsunami*.

In this regard, the term *flood* (Indonesian: *banjir*) is unlikely to evoke the real danger that would be caused by a dyke-break event, since local inhabitants may imagine a gradually increasing persistent inundation, particularly in Jakarta. Based on this consideration, we coined *dyke-break induced tsunami* to get people to easily imagine the serious consequences that could arise from the break of a dyke. Also, we expect that the usage of this term may be acceptable, as *dam-break induced tsunami* or *landslide-induced tsunami* are similarly used to describe the danger caused by a sudden movement of a large water mass. The *dyke-break induced tsunami* is considered similar to those expressions. Nevertheless, we do understand the concern raised by the reviewer. Thus, we will clarify this point more carefully in the revised manuscript.

2. It would be advisable that the authors mentioned sea water desalination as a counter action potential solution against land subsidence induced by underground water withdrawal.

We agree. A recent NHESS article by Budiyono et al. (2016) describes so-called "100-0-100" sanitation policy issued by the Ministry of Public Works (PU), Indonesia. We will refer to their paper to emphasize the importance of mitigating underground water withdrawal in the revised manuscript.

Budiyono, Y., Aerts, J. C. J. H., Tollenaar, D., Ward, P. J. (2016) River flood risk in Jakarta under scenarios of future change, Nat. Hazards Earth Syst. Sci., 16, 757-774, DOI: 10.5194/nhess-16-757-2016

3. A fast and significant subsidence rate has been indicated for the recent past years. It is not clear on what basis the same rate is maintained for the coming 10 years as well as for further time states. The subsidence would depend on the soil type of the underground and the thickness of the pervious layers, so it is not necessary correct to extrapolate the same sinking rate for the

future, unless the pervious soil and its thickness give base to this assumption, fact that is not stated.

As the reviewer suggests, the land subsidence appears to be the most difficult issue in projecting floods in a rapidly developing coast such as Jakarta. Since publicly available data is very limited in the country, it may not be easy to project the subsidence rate by a theoretical method (*e.g.* Terzaghi consolidation theory). Thus, we simply assumed that land subsidence will continue at the present rate for a while (at least 10-15 years), resulting in a 2-meter subsidence in the studied area, based on the subsidence rate over the last couple of year shown in Fig.1 and other published articles on subsidence in Jakarta. We consider this projection does not really overestimate things, given the recent rate continues unabated at present.

[Figure]

Figure 1: Land subsidence in Asian megacities

4. It is not clear also if the plantation of mangrove forest will be able to provide the expected protection in future. The present water depth in the proposed plantation area is indicated as 50 cm and that the plants grow at approximately as the present sea level rise. However, new publications (e.g. Dutton et al., 2015; De Conto and Pollard, 2016; Hansen et al., 2015; Mengel et al., 2015;) indicate a feasible faster and larger sea level rise globally (up to 1m by 2050, 2-3 m by 2100), in which case the mangrove will not be able to grow at the same rates and provide

the expected protection from dyke-break rapid flooding. Perhaps an engineering alternative could be to adopt the Dutch concept, of building wide sand dunes at the waterline, requiring resettling of population and its activities back to higher and more remote places from the lower sea/water front areas. Another theoretical option might be migration to higher places (Roberts and Andrei, 2015) or that adopted by Miami City in USA (Weiss, 2016).

Thank you very much for raising this very important issue associated with the accelerated pace of SLR. The embankment will end up being submerged under mean sea level, resulting in the loss of its function against potential coastal floods. We will refer to those articles suggested by the reviewer and discuss engineering issues to be solved for creating the mangrove forest and these alternative solutions in the revised manuscript.

5. The criteria proposed following the classification given by Pistrika et al. as well as the one proposed by Wright et al., (2010) seem very problematic as explained further below. Also the data brought by Suga et al., 1995) indicating a safe velocity limit of up to 0.8 m/s in a water depth of about 0.8 m, whereas at speed of 1 m/s was the highest safe limit walking against the current. Based on a research beach bathers survey study carried in Japan with in order to determine safe recreational conditions, that paper stated a safe limit of current speed of about 0.15m/s for knees deep water flow (about 0.5m depth), beyond which bathers could not walk normally or remain stable. Unfortunately, I was not able to find this article published in the "Coastal Engineering in Japan", in the 1980's. In the present article, the authors selected to use a depth velocity product criterion to determine safe passage of pedestrians in a flooded area. The present paper describes a flooding in the Philippines where people could cross a flooded street in a water depth of 0.6 m (knees depth) and while a 0.6 m/s current flow was present. The information provided about the persons particulars is very limited and about which type of street (paved, unpaved, etc) was crossed in the flooded area. This seems already dubious as it is not clear how one was able at the time to measure the current speed, which, if it was 0.6 m/s (based on the Japanesse paper I mentioned), should have been done by a tall, heavy and strong person with perhaps even some cable support from being carried away. A velocity-depth product of 1.0 $m^2$/s seems already unsafe, if we consider that this product can be due to various scenarios, such as: a depth of 1m and speed of 1.0 m/s (2 knots); a 1.8 water depth in a 0.55 m/s current (1 knot); or a water depth of 0.6m in a current speed of 1.67 m/s. These all lead to same velocity-depth product of 1 $m^2$/s. A more rigorous approach is the work of Cox et al., 2010, quoted by Pistrika et al., which in this reviewer's opinion is a very important one. We believe it would be appropriate that the authors quote the following text taken from Cox and al., 2010, or

at least refer to it and give a summarizing figure from that publication, copied further below as Figure 1. Since the Cox et al. report is more recent and of broader coverage, and since it refers in greater detail to the various types of persons and ages and floor bottom conditions, even if the Japanese article was right for the wave induced current under open coast conditions with waves and sandy sea bottom, we estimate using the Cox et al. report would be more adequate for use in the present article.

We thank the reviewer for reminding us to those important work which we didn't quote in the first draft of our manuscript Although we consider the report of Wright and his collaborators provides meaningful insights for safety criteria during floods, we do also agree that the criteria cannot be represented by only one single value, given uncertainties associated with various factors. The reviewer's suggested article seems to give a more comprehensive and conservative criteria by taking into account *e.g.* a height and mass product, which enables to understand what would happen to either children or adults. In the next revision, we will also introduce reference to Cox et al. (2010) in a way such that:

"Wright et al. (2010) proposes a depth-velocity product of $1.0 m^2/s$ as the safe limit for pedestrians. *However, a plot of the relationship between human's instability and flow regime appears to be scattered by multiple factors such as surface material; subject actions -either standing or moving-, experience and training, clothing and footwear and physical attributes including muscular development and/or other disabilities; the definition of stability limit (i.e. feeling unsafe or complete loss of footing). Thus, the depth-velocity product criteria suggested by Wright et al. (2010) could become optimistic for some adverse conditions. Regarding physical differences between adult and child, Cox et al. (2010) suggests that for children with a height and mass product of between 25 and 50, low hazard exists for flow values of the depth-velocity product < 0.4 $m^2/s$, with a maximum flow depth of 0.5 m regardless of velocity and a maximum velocity of 3.0 $m^2/s$ at shallow depths (D < 0.2 m).*"

b. Table 1 List of technical and typographical corrections suggestions to the article content Fig. 2

We will reply to those suggestions listed on the table in the revised manuscript.

---

## Author Comment (AC2) · 26 Jun 2016

In blue: reviewers' comments

In red: authors' reply

Anonymous Referee #2

Revision comments to the paper: Mangrove Forest against Dyke-break induced Tsunami in Rapidly Subsiding Coasts. H. Takagi, T. Mikami, D. Fujii1, M. Esteban

The paper is well written and documented, but there are a few misspellings

| Page | Line | Comment |
|------|------|---------|
| 2 | 5 | Correct abstraction by extraction |
| 2 | 23 | Correct dyke by dykes |
| 3 | 26 | Correct m/s by m3/s |
| 6 | 1 | Correct Losada by Losada et al. |
| 6 | 9 | Change "resembling which resembles" by "like" |
| 9 | 21 | Reference Kaneko S., Toyota T. is not commented in the text |
| 10 | 12 | Reference Reed D.J. is not commented in the text |

We thank the reviewer for a number of very productive comments and suggestions, which would enable us to significantly improve our manuscript. All of the comments above will be modified accordingly.

Figure 7 showing the models bathymetry indicate a steady topography descent from the upper part of the domain (where the dyke breach is located) to the opposite side. Despite the poor selection of colours it can be seem that the roads around the breach are at around -1.5 m, while point 7 and beyond in the other side of the domain is about -3 m or more. This is not congruent with the much complex measured topography Printer-friendly version Discussion paper indicated in figure 3.

As the reviewer points out, the lower half of the plots indicated in Fig. 3 (c) shows a ring road which is relatively higher (-1.0 – -1.5 m) than the lowest part of the town (-2.5 – -3.0 m). However, the authors came to realize that the road was higher than surrounding areas. There were some corners where the ground measurement was not possible because of the difficulties in access to the area due to the presence of an off-limits area or informal settlement (please see the photo below). Instead of using the elevation along the road, the elevation of the lower half of the domain was conservatively assumed to be the same as the lowest part of the town in the numerical

model. The details on this treatment will be described in the revised manuscript.

[Figure]

The velocity figures for the present scenario without mangrove protection and in all cases with it show that by the simulation end (20 minutes) water is flowing at steady pace in all points. As water depth is also stabilised, where is the water flowing? There are open boundaries? There is insufficient data on the paper about the boundaries of the numerical domain, so results are of difficult interpretation based on the presented information.
A better figure 7, showing clearly the model boundaries and the topography will be welcomed.

Since the simulation runs for only the beginning of 20 minutes after the breach, sea water passes through the community and accumulates in a lower part of the computational domain, making it possible to avoid imposing pseudo-open boundaries on the land. In the revised manuscript, the details will be carefully clarified.

The wild oscillations of water level and velocity shown in the case of 2 m subsidence case without mangrove protection are not commented. Could it be related to model instabilities?

We agree. The choppy flood state in the case of 2m subsidence scenario will also be described in the revised manuscript.

In relation to the capacity of people to withstand a water flow of a given depth and velocity, the

proposition of Wright et al. (2010) of a depth-velocity product of 1.0 m$^2$/s as the safe limit for pedestrians seems optimistic. On that respect, Jonkman, S.N. and Penning-Rowsell, E. (2008) proposed some formulas for both moment and sliding instability of pedestrians, depending on the individual mass, friction factor and flow depth and velocity. Using these formulas, this 1 m$^2$/s limit seems only valid for trained adults. A more recent paper of Cox et al. (2010) provides a much more realistic table about the safe limits for wadding in water flows for people.

In the next revision, we will also introduce reference to Cox et al. (2010) in a way such that:

"Wright et al. (2010) proposes a depth-velocity product of 1.0m$^2$/s as the safe limit for pedestrians. *However, a plot of the relationship between human's instability and flow regime appears to be scattered by multiple factors such as surface material; subject actions -either standing or moving-, experience and training, clothing and footwear and physical attributes including muscular development and/or other disabilities; the definition of stability limit (i.e. feeling unsafe or complete loss of footing). Thus, the depth-velocity product criteria suggested by Wright et al. (2010) could become optimistic for some adverse conditions. Regarding physical differences between adult and child, Cox et al. (2010) suggests that for children with a height and mass product of between 25 and 50, low hazard exists for flow values of the depth-velocity product < 0.4 m$^2$/s, with a maximum flow depth of 0.5 m regardless of velocity and a maximum velocity of 3.0 m$^2$/s at shallow depths (D < 0.2 m).*"

We will also quote the paper of Jonkman and Penning-Rowsell (2008), which emphasizes that friction instability appears to occur earlier than moment instability (toppling) for the combination of shallow depth and high velocities.

Jonkman, S.N., Penning-Rowsell, E., 2008. Human instability in flood flows. J. Am. Water Resour. Assoc. 44 (4), 1–11.